# Compound flood impact of water level and rainfall during tropical cyclone period in a coastal city: The case of Shanghai

Hanqing Xu[1,2,3], Zhan Tian[2], Laixiang Sun[4,5], Qinghua Ye[3,6], Elisa Ragno[3], Jeremy Bricker[3,7], Ganquan Mao[2], Jinkai Tan[8], Jun Wang[1], Qian Ke[3], Shuai Wang[9], Ralf Toumi[9]

[1]Key Laboratory of Geographic Information Science (Ministry of Education), East China Normal University, Shanghai, China
[2]School of Environmental Science and Engineering, Southern University of Science and Technology, Shenzhen, China
[3]Department of Hydraulic Engineering, Faculty of Civil Engineering and Geosciences, University of Technology, Delft, Netherlands
[4]School of Finance & Management, SOAS University of London, WC1H 0XG London, UK
[5]Department of Geographical Sciences, University of Maryland, College Park, USA
[6]Deltares, Delft, Netherlands
[7]Department of Civil and Environmental Engineering, University of Michigan, Ann Arbor, Michigan, USA
[8]School of Atmospheric Sciences, and Key Laboratory of Tropical Atmosphere-Ocean System (Ministry of Education), Sun Yat-sen University, Zhuhai, China
[9]Department of Physics, Imperial College London, London, UK

*Correspondence to*: Zhan Tian (tianz@sustech.edu.cn); Jun Wang (jwang@geo.ecnu.edu.cn)

**Abstract.** Compound flooding is generated when two or more flood drivers occur simultaneously or in close succession. Multiple drivers can amplify each other and lead to greater impacts than when they occur in isolation. A better understanding of the interdependence between flood drivers will facilitate a more accurate assessment of compound flood risk in the coastal regions. This study employed the D-Flow Flexible Mesh model to simulate the historical peak coastal water level, consisting of storm surge, astronomical tide, and the relative sea level rise (RSLR) in Shanghai over 1961-2018. It then applies a copula-based methodology to calculate the joint probability of peak water level and rainfall during historical tropical cyclones (TCs) and to calculate the marginal contribution of each driver. The results indicate that the astronomic tide is the leading driver to peak water level, followed by the contribution of storm surge. In a longer term, the RSLR has significantly amplified the peak water level. This study investigates the dependency of compound flood events in Shanghai on multiple drivers, which helps us better understand compound floods and provides scientific references for flood risk management and for further studies. The framework developed in this study could be applied to other coastal cities which face the similar constraint of unavailable water level records.

## 1 Introduction

Compound flooding is generated when two or more flood drivers, e.g., water level, rainfall, and high river discharges, occur simultaneously or in close succession. Such flood drivers can amplify each other and lead to greater impacts than when they occur in isolation (Leonard et al., 2014; Zscheischler et al., 2018; Visser-Quinn et al., 2019; Chao et al., 2021). Coastal cities

like Shanghai are particularly prone to compound flooding associated with tropical cyclones (TCs), which often bring heavy rainfall and storm surge. For a more accurate assessment of compound flood in the coastal regions, a thorough understanding of the interdependence between multiple flood-drivers is necessary. In other words, an enriched knowledge about the dynamic interaction between flood-drivers would significantly improve the quantification of compound flooding risks in estuarine environments (Feng and Beighley, 2020). As such, the joint probability theory has been incorporated into the analysis of compound flood risk to take the advantage of the Sklar's Theorem (M. Sklar, 1959). According to Sklar's Theorem, any multivariate joint cumulative distribution function can be expressed in terms of univariate marginal distribution functions and a copula which describes the structure of dependency between the variables (Bevacqua et al., 2019).

Coastal regions are usually the most densely populated and economically developed areas of a country, and they are also the most vulnerable regions to the risk of compound flooding from heavy rainfall and extreme storm surge due to this large population and property density (Neumann et al., 2015). Shanghai is the largest and most developed coastal megacity in China. Rainstorm and storm surge caused by typhoon from June to October often cause substantial losses (Li et al., 2018; Yin et al., 2021). For example, extreme storm flooding caused nearly 30 thousand casualties in 1905 (Li et al., 2019). In 1962, storm flooding inundated half of the downtown region for nearly 10 days due to 46 failures along the floodwalls of the Huangpu River and its branches and led to huge losses of 1/6 of the local Gross Domestic Product (GDP) in Shanghai (Ke, 2014). In 1997, Typhoon Winnie killed seven people and flooded more than 5,000 households due to the extreme storm surge and rainfall (Ke et al., 2021). Although the construction of flood control measures in the past 50 years (especially after typhoon Winnie in 1997) has effectively reduced the risk of storm surge and rainstorm floods, Typhoon Matsa in 2005 (US $2.23 billion damage), Typhoon Fitow in 2013 (US $10.4 billion damage), and Typhoon Lekima in 2019 (US $2.55 billion damage) also brought significant damage to Shanghai (Du et al., 2020). Given the substantial damage caused by compound flooding, comparing the encounters of rainfall and storm surge during typhoon season is urgent in order to understand the driving mechanisms and frequency of compound flooding in Shanghai. However, owing to the unavailability of water level records, there is little research that has been able to estimate the dependency between peak water level and accumulated rainfall during historical TCs.

The copula method is widely used in statistics to model the interdependence between two or more variables (Anandalekshmi et al., 2019; Balistrocchi et al., 2019; Xu, P. et al., 2019). Recent research using the copula model emphasizes the importance of studying the combined effects of rainfall and water level processes in estuaries and coastal regions (Zheng et al., 2013; Wahl et al., 2015; Zellou and Rahali, 2019). For example, Xu, H. et al. (2018) showed the existence of some positive dependences between rainfall and water level in a coastal city of Hainan Island, while the water level poses an additional risk of flooding. The studies of both Xu, P. et al (2019) and Xu. H. et al. (2018) confirmed that the copula method is a promising tool for studying multivariate problems in hydrology and coastal engineering. However, when applying the copula-based methods to 3 dimensions, controversies arise and uncertainty can become explosive (Bevacqua et al., 2017; Santos et al.,

2021). The univariate flood driver cannot provide an accurate evaluation if the underlying drivers are modelled as independent extreme events (Li et al., 2016; Khanal et al., 2019).

Flood induced by TCs is the most frequent natural disaster in the eastern coastal region of China (Zhang et al., 2020). The East Asian typhoon season is characterized by heavy inland rainfall and high storm tide, which are the major driving factors of coastal flood hazards in China. The slowdown in forward speed of landfalling TCs in the Northwest Pacific over 1949-2015 had increased the risk of flooding from water level and rainfall even without considering the changes in storm strength (Kossin, 2018). The simultaneous and/or consecutive occurrence, both in time and space, of heavy rainfall and high tide can lead to compound flooding (Wahl et al., 2015; Bilskie et al., 2021; Liu et al., 2022). Furthermore, the risk posed by the interactions between hydro-meteorological events under the condition of sea level rise and changing tidal regimes is bound to increase in the future (Idier et al., 2020). Despite the increasing threat of compound flooding events along the Chinese coast, owing to the unavailability of water level records during typhoon events, the associated joint probabilities and driving mechanisms have not been explored (Fang et al., 2021). This research intends to fill this important niche.

The TCs often produce strong onshore winds and low barometric pressure, which would cause extreme storm surge, at the same time, generate heavy rainfall on the coastal region (Hoque et al., 2018; Sohn et al., 2021). Peak water level during TCs not only results from the combination of storm surge and astronomical tide. Additionally, the combination of absolute sea level rise (SLR) due to the global warming and land subsidence due to urbanization has caused relative sea level rise (RSLR) (IPCC, 2021; Jebbad et al., 2022). According to the Regulations of Shanghai Municipality on the Administration of Land Subsidence Prevention and Control, the land subsidence rate was 6.19 mm/yr from 1965 to 2001. Since 2001, the land subsidence rate has been controlled to varying degrees by adaptation measures such as recharging water to aquifers.

This study establishes the joint distribution of peak water level and rainfall during typhoon events in the Shanghai estuary region, with the aim to better understand the risk of compound flooding and to improve the assessment of flood-defence design standard for adaptation strategies. Our modeling framework couples a state-of-the-art hydrodynamic model and statistic model. This model coupling enables us to quantify the joint distribution of rainfall and storm surge events during typhoon season, and also to consider the comparative cases with and without RSLR for Shanghai. The procedure of the modeling framework is as follows. First, the peak water levels, consisting of astronomical tides, storm surges associated with TCs, and RSLR, in Shanghai over 1961-2018 are generated using the D-Flow Flexible Mesh (D-Flow FM) model, then a compound hazard scenario for deriving design values is chosen. Second, we compare and investigate the peak water level with and without RSLR, and select the extreme compound flood events according to the design standard of the joint hazard scenario. Finally, we analyse the contribution of storm surge, astronomical tide and RSLR to peak water level based on the top seven extreme compound flood events over the study period. We provide a framework that could be applied to other coastal cities which face the similar constraint of unavailable water level records. The findings from our research could be useful for decision-makers in developing coastal flood defence measures in Shanghai and other East Asian coastal cities. This is the major contribution of this research.

## 2 Materials and Methods

### 2.1 Study Area

Shanghai is surrounded by water on three sides, and the Huangpu River and Suzhou River pass through the city (Figure 1). The total area of Shanghai is 6,340.5 km$^2$ with a population of 24.87 million in 2020. The annual rainfall is around 1,200 mm. June to September are the rainy months. From late August till early September, Shanghai is frequently affected by typhoons and rainstorms (Yin et al., 2021). Storm flooding caused by typhoons is the main natural disaster in Shanghai. Shanghai's flood risk is about US $63 million/year under an optimistic scenario of a maximum protection level of 1/1000 per year (Hallegatte et al., 2013). Although the construction of flood control measures in the past 50 years has effectively reduced the risk of storm floods, TC Matsa in 2005, the 2013 TC Fitow, and the 2019 TC Lekima caused substantial losses in Shanghai. Particularly, typhon Winnie in 1997 led to a direct economic damage of over US $100 million. During typhoon Winnie period, the peak water level at Huangpu Park (city center) rose to 5.72 m, equivalent to the water level with a 500-year return period. During typhon Fitow in 2013, the water level at Mishidu in the inland area of the Huangpu River was recorded at WD (Wusong Datum is adopted as the reference) as 4.61 m, which broke the record (Ke et al., 2018). In the context of climate change, relative sea level rise, and urban expansion, Shanghai will face higher compound flood risks and challenges from TCs, storm surge, and extreme rainstorm in the future (Wang et al., 2018).

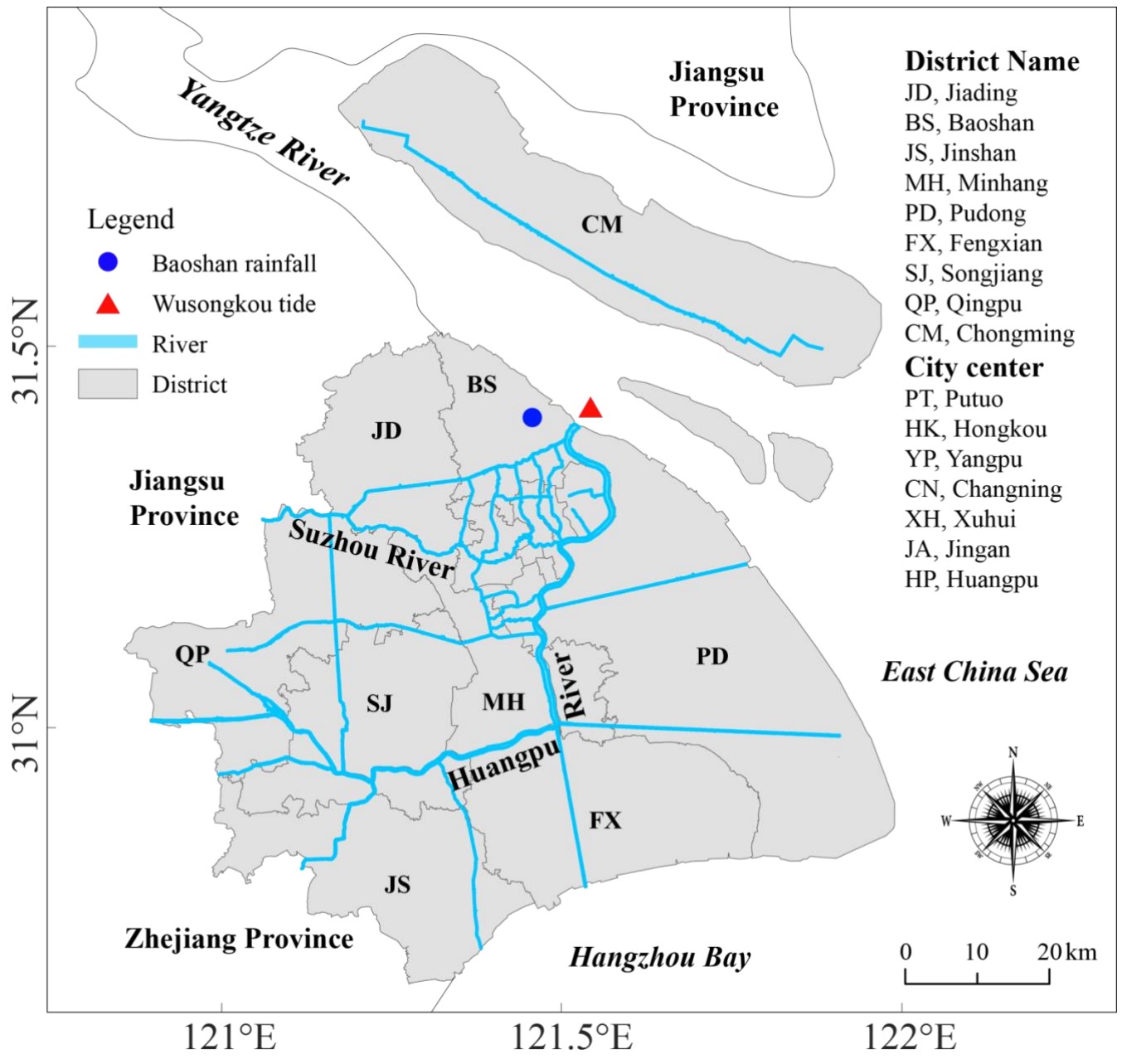

**Figure 1. Location map of Shanghai**

**2.2 Data**

This study systematically collected the geographic and meteorological data of the study area, including TC tracks (1961-2018), and daily accumulated rainfall (1961-2018). Due to the unavailability of measured water level data from historical TCs, in this study we evaluate the dependence coefficient between rainfall and water level based on observed rainfall and

simulated peak water level during TCs. Observations come from a set of rain gauge measurements. Modelled data consist of data sets created using the International Best Track Archive for Climate Stewardship (IBTrACS) from NOAA's National Climatic Data Center.

We use the D-Flow FM model (Knapp et al., 2010) to simulate water level during TC periods. IBTrACS contains 6-hourly TC centre's longitude and latitude, minimum central pressure ($P_c$) and sustained maximum surface wind velocity ($V_{max}$). Multiple agencies provide TC best tracks in the West Pacific, and we opt to use the best track from Hong Kong Observatory (www.hko.gov.hk). This choice was made because it includes the most complete set of observations from TCs making landfall in China (Chen et al., 2011).

We analysed the historical TCs influencing Shanghai between 1961 and 2018. We first defined a 6-degree-latitude square box around Shanghai (Figure 2). The area covered by the blue box can be regarded as an alert area in terms of a TC causing potential damage in Shanghai. The size of the blue box was designed to be just large enough to include the partial tracks of the top 10 most severe TCs for Shanghai since 1949 (personal communication with Shanghai Climate Centre). We then selected historical TCs lasting for at least 24 hours in the blue box. After this best-track pre-processing, 210 TCs for the

period of 1961-2018 are selected in this study (Figure 2). Additionally, we obtained tidal level data (1997) at the Wusongkou tide gauge from the Shanghai Municipal Water Authority, which are used for hydrodynamic model validation.

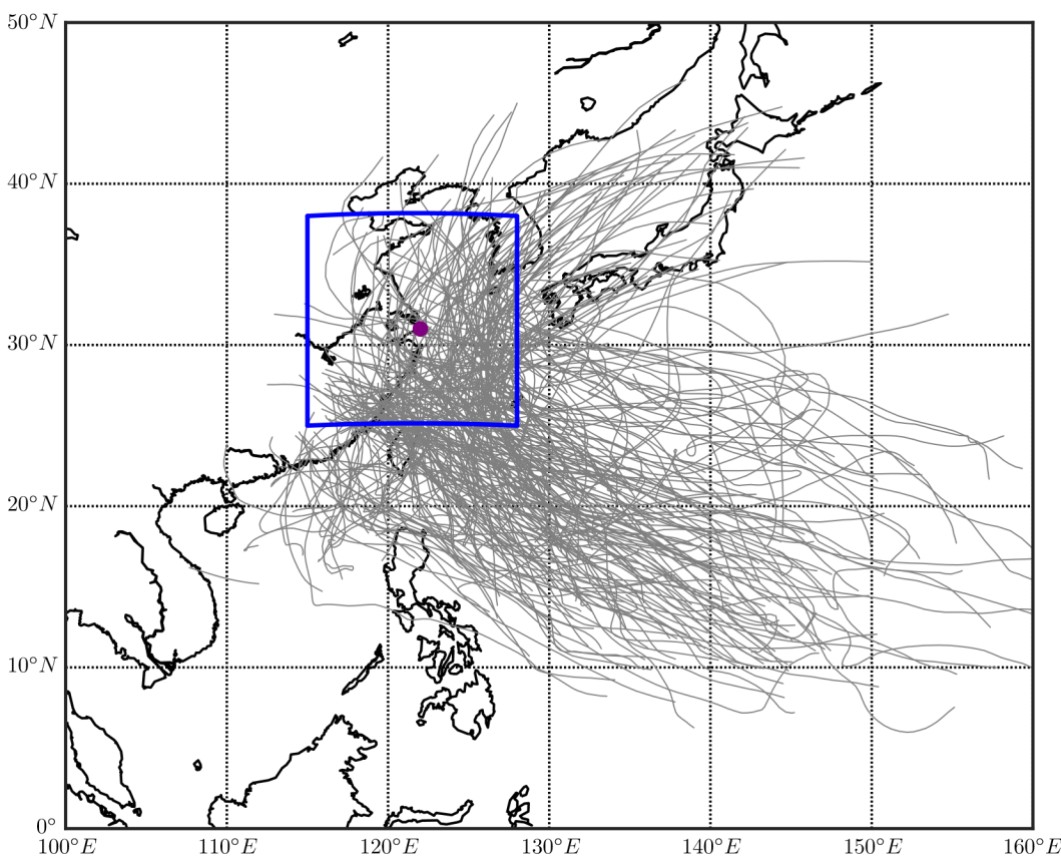

**Figure 2. Location map for the area of interest. (Grey colored lines indicate major historical typhoon tracks within the region. Blue box indicates the selection criteria.)**

Daily rainfall records from 1961 to 2018 are collected from the China Meteorological Administration (CMA, http://data.cma.cn) for the Baoshan gauge station, being the closest to the Wusongkou surge station (Figure 1). The annual precipitation in Shanghai is 1,200 mm with the rainiest months being from June through September. Rainfall data are used in this study to approximate the TC-induced runoff. To implicitly account for the rainfall travel time to the catchment outlet, 1-, 2-, and 3-days accumulated rainfall were also estimated and the correlation between such accumulated rainfall and peak

water level was then estimated.

According to the Chinese Sea Level Bulletin of 2020, which was compiled by the State Oceanic Administration of China, the absolute sea-level rose at a rate of 3.4 mm/yr. According to the Regulations of Shanghai Municipality on the Administration of Land Subsidence Prevention and Control, the land subsidence rate was 27.93 mm/yr from 1921 to 1964. From 1965 to 2001, the land subsidence rate was 6.19 mm/yr. After 2001, the land subsidence rate has been under varying

extents of control by adaptation measures such as recharging water to aquifers, and in most regions being 5-15 mm/yr. We use 10 mm/yr as the land subsidence rate from 2001 to 2018. The downside of such an assumption is that it fails to consider possible accelerating factors such as population growth, vertical and horizontal urban expansion, and deep strata motions, but these complex factors are beyond the scope of this research.

## 2.3 The framework

The objectives of this study are to overcome the limitation of unavailable water level records during TCs and set up a framework to improve the methods for selecting the most suitable TCs for the research and for investigating TCs' influence on water level. Due to the limited water level data availability, we employ an empirical track model for pressure and wind fields, followed by a physics-based ocean model to simulate storm tide and astronomical tide during TCs in Shanghai. A numerical simulation is carried out to better understand the distribution and timing of the peak water level and the areas of

the country affected. The physics-based ocean model was calibrated using the recorded atmospheric pressure and focused on ones with the most severe damages, comparing well with the results of the field survey data (Ke et al., 2021). Following this, the copula function was used to connect peak water level with accumulated rainfall and construct a joint distribution. After that, we compare and investigate difference between peak water level and accumulated rainfall under the effect of RSLR, and select the extreme compound flood events according to the design value of the joint hazard scenario. Finally, we analyse

the contribution of storm surge, astronomical tide and RSLR to peak water level for the extreme compound flood events (Figure 3).

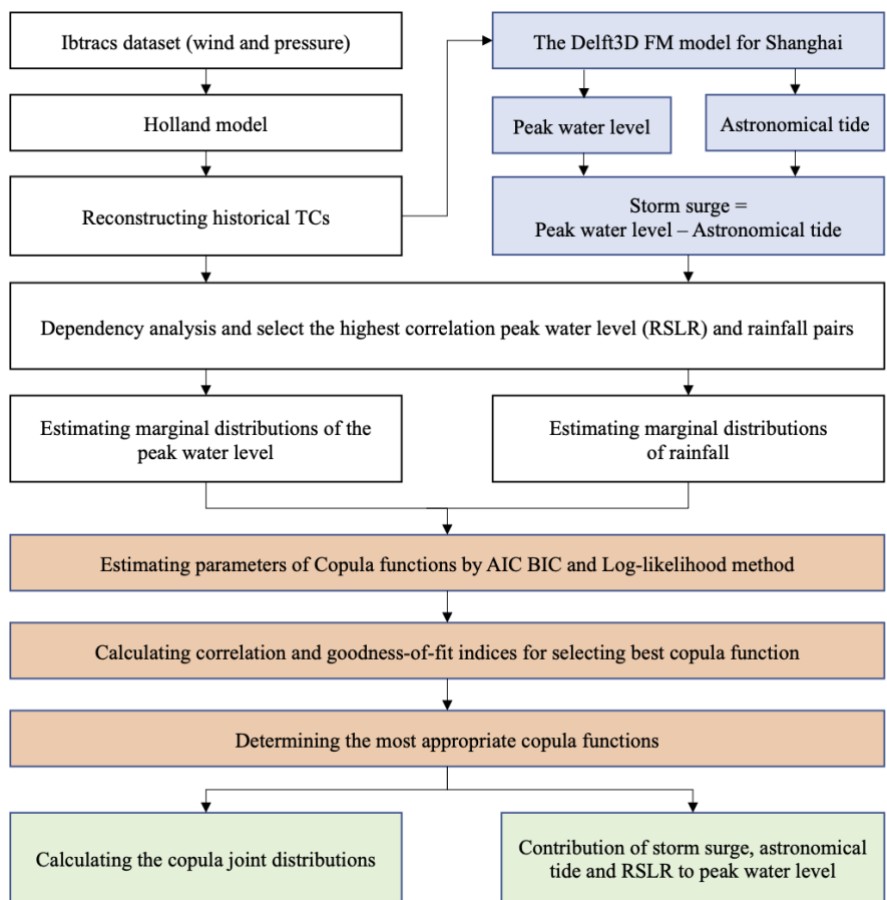

**Figure 3. Flowchart of this study**

### 2.4 D-Flow FM model

Delft3D WES (Wind Enhance Scheme), a built-in module in Delft3D, is used to generate wind and air pressure fields of each TC according to the Holland formula (Holland et al., 2010). It is able to generate tropical cyclone wind and pressure fields around storm center positions on a high-resolution grid. Delft3D WES slightly improves on this by including asymmetry. This asymmetry is included by the use of the translational speed of the cyclone center's displacement by the steering flow, and the rotation of the wind velocity due to friction (Takagi and Wu, 2016). The output of Delft3D WES is

suitable as input for the D-Flow FM model to simulate water level including the effect of storm surge.

   The D-Flow FM module as part of the Delft3D Flexible Mesh suite solves the non-linear shallow water equations for unsteady flow and transport phenomena derived from the three-dimensional Navier Stokes equations for incompressible free surface flow (Symonds et al., 2016). The hydrodynamic model D-Flow FM is employed in this study to solve multi-disciplinary problems in coastal, river, and estuarine environments (Deltares, 2013). The domain of the model covers the

East China Sea, Hangzhou Bay, the Yangtze Estuary, and the downstream reach of the Yangtze River, ranging from 24 to

34°N and 118 to 128°E, and consists of 69,000 mesh cells. The model has been validated with observed storm tide and astronomical tide at 10 stations around Shanghai during TC Winnie in 1997 (Ke et al., 2021). The storm tide and astronomical tide of 210 TCs are calculated in this D-Flow FM model. Then, the peak storm tide is selected from each TC. In addition, the storm surge is calculated by using storm tide minus astronomical tide at the same time of peak storm tide. In this study, we assume the Yangtze River discharge equals its annual mean at 31000 m³/s.

## 2.5 Dependence modelling and design value via copulas

We define the joint distribution of accumulated rainfall and peak water level, $F_{R,WL}$ as $F_{R,WL} = C(F_R, F_{WL})$ where $F_R$ and $F_{WL}$ are marginal distributions of accumulated rainfall and peak water level, and $C$ is the associated dependence function, i.e., copula, modelling the dependence between accumulated rainfall and peak water level independently from their marginal distributions (Sklar, 1973; Salvadori and De Michele, 2004). Hence, we select marginal distributions among the most commonly used distribution functions for extremes, namely: Generalized Extreme Value (GEV), Pearson type III (P-III), Gumbel, Exponential and Weibull.

The copula function raised by Sklar can model the dependence structure and joint probability distributions. The Gaussian, Clayton, Frank and Gumbel copula functions are selected to establish joint distribution between peak water level and accumulated rainfall. To evaluate the fitting error and select the appropriate copula function by the non-parametric estimation method, the Akaike information criterion (AIC), Bayesian information criterion (BIC) and root mean square error (RMSE) are employed.

$$AIC = -2\iota(\hat{\theta}|y) + 2K, \tag{1}$$

$$BIC = -2\iota(\hat{\theta}|y) + K ln(n), \tag{2}$$

$K$ is the number of estimated parameters in the model including the intercept and $\iota(\hat{\theta}|y)$ is the log-likelihood at its maximum point of the estimated model; $n$ is the sample size. The rule of selection was that the smaller the value of AIC was, the better the model was, and similarly with the BIC.

$$RMSE = \sqrt{\frac{1}{n}\sum_{i=1}^{n}\left(X_C(i) - X_O(I)\right)^2}, \tag{3}$$

where $n$ is the number of observations; $X_C$ is the theoretical probability from the copula and $X_O$ is the empirical observed probability. It is also worth noting that the dependence between accumulated rainfall and peak water level is given by their linear correlation, i.e., Spearman's $\rho$, or concordant/discordant pairs, i.e. Kendall $\tau$.

Following Salvadori and De Michele (2004), copulas allow a straightforward definition of two hazard scenarios, i.e, pairs with an occurrence probability greater than a safety threshold, namely "AND" and "OR" scenarios. The "AND" scenario assumes that a hazardous condition is realized when both the dependent variables, in this case rainfall and water level, exceed their predefined thresholds, while the "OR" scenario assumes that a hazardous condition can occur when either one

of the two dependent variables exceed their predefined thresholds. The "AND" scenario is commonly used for compound flooding mostly because the flooding can be generated by excessive runoff, high sea level, or a combination of both (Moftakhari et al., 2017; Moftakhari et al., 2019; Zellou and Rahali, 2019). The joint exceedance probability based on the "AND" scenario is given by Eq. (4).

215 $$P\big((U > ud) \cap (V > vd)\big) = 1 + u_d + v_d - C(u_d, v_d)\,, \tag{4}$$

where $U = F_R$ and $V = F_{WL}$ are the marginal distributions and $u_d$ and $v_d$ are the safety threshold of accumulated rainfall and peak water level, respectively. The dependent design values $\big(R_d = (F_r - 1)(u_d), WL_d = (F_{WL} - 1)(v_d)\big)$ can be inferred from Eq. (4) based on the level of safety desired.

The joint probability cannot be directly used as the reference value of the actual engineering fortification standard. We calculate the joint design value combinations with the joint return period, which can serve as a reference for the engineering design. For given peak water level and accumulated rainfall events, under the conditions of a given joint return period, we design a series of $(u_d, v_d)$ combinations to maximize $P((U > u_d) \cap (V > v_d))$, thereby obtaining the optimal combination design value. In the practical calculation, the intersection of the diagonal of critical probabilistic surface and probability isoline is regarded as the design values of $(u_d, v_d)$.

## 3 Results

### 3.1 Effect of RSRL to peak water level

The correlation between peak water level and accumulated rainfall is positive (Table S1). The peak water level and 2-days accumulated rainfall have the highest correlation compared with 1-days and 3-days accumulated rainfall. The correlation between peak water level and accumulated rainfall is significant ($P_{value} < 0.05$) in all cases. Consequently, the remaining analysis will be performed considering 2-days accumulated rainfall, here after $R_{2d}$.

Probability density function is a useful tool for comparing peak water levels between the cases with and without RSLR. Results in Figure 4 shows a clear shift in the distribution of peak water level during the TC periods. It demonstrates that RSLR increases both the mean and variance of peak water levels, thus resulting in higher risk of flooding in Shanghai.

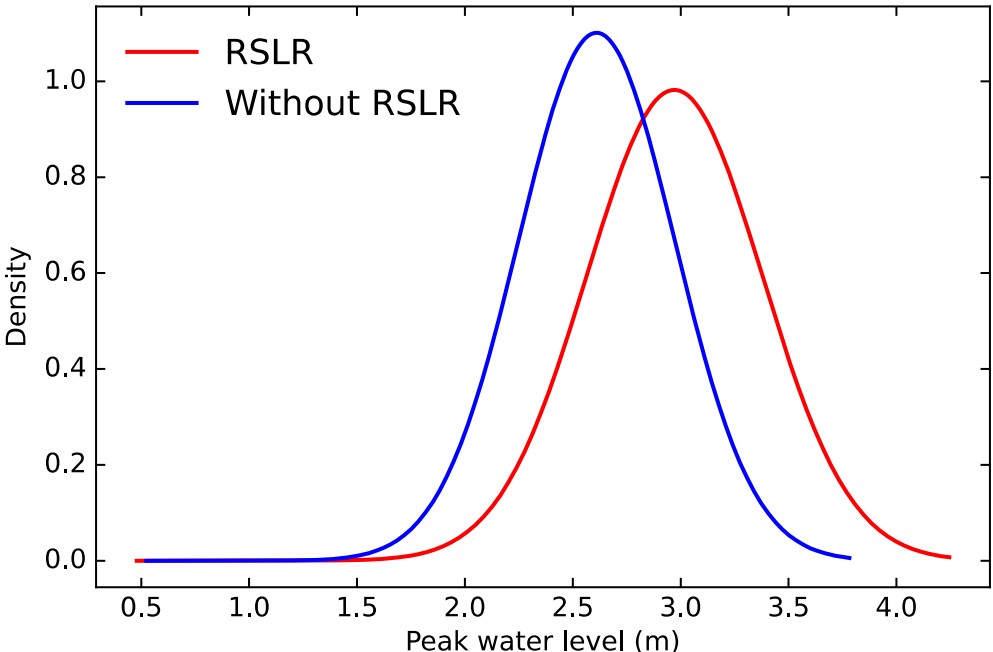

**Figure 4. The shift of the probabilistic distribution of peak water level, "with RSLR" versus "without RSLR", in Shanghai during 1961-2018.**

The marginal distributions are used to transform peak water level and $R_{2d}$ into uniform marginals, $u_{WL}$ and $u_{R2d}$, respectively. Then, the preferred copula is selected among the Clayton, Frank, Gumbel and Gaussian copula. In the case without considering RSLR, the preferred copula is Clayton because it has the smallest AIC, BIC, and RMSE (the upper panel in Table 1). In the case with the presence of RSLR, the lower panel in Table 1 shows that Gaussian copula has the smallest AIC, BIC, and RMSE. Therefore, Gaussian copula is selected as the best fit for the peak water level and accumulated rainfall under the effect of RSLR. Figure 5 shows the difference between peak water level and accumulated rainfall with RSLR and without RSLR. This indicates that different copula families can return different dependence structures. In Figure 5, both peak water level and accumulated rainfall are presented in probability space. Gaussian and Clayton copula families are used to explain the bivariate dependence between peak water level and accumulated rainfall in this study. The red and blue isolines are fitted Gaussian copulas and Clayton copulas, respectively. Neither is among the commonly used copulas in the hydrological literature. This highlights the importance of the choice of the copula, and quantifies the difference in results based on copula choice.

**Table 1. Performance measures of the estimated copula functions**

| | Copula type | Max-likelihood | AIC | BIC | RMSE |
|---|---|---|---|---|---|
| Without *RSLR* | Gaussian | 1024.1 | -2046.1 | -2042.8 | 0.1105 |
| | **Clayton** | **1034.7** | **-2067.5** | **-2064.1** | **0.1050** |

| | | | | | |
|---|---|---|---|---|---|
| | Frank | 1009.4 | -2016.8 | -2013.4 | 0.1185 |
| | Gumbel | 972.0 | -1941.1 | -1938.7 | 0.1415 |
| | **Gaussian** | **1038.8** | **-2075.5** | **-2072.2** | **0.1030** |
| Presence of *RSLR* | Clayton | 1029.4 | -2056.7 | -2053.4 | 0.1077 |
| | Frank | 1016.4 | -2030.9 | -2027.5 | 0.1146 |
| | Gumbel | 992.7 | -1983.5 | -1980.2 | 0.1282 |

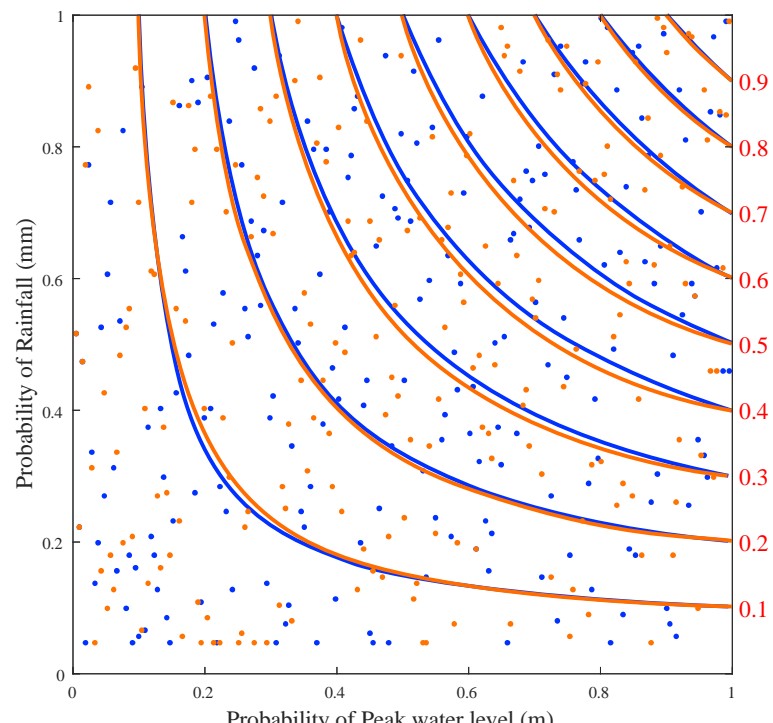

**Figure 5. With RSLR (red) and without RSLR (blue) for 2 different copulas. Both peak water level (x axis) and accumulated rainfall (y axis) are presented in probability space. The red isolines are the fitted Gaussian copula, and the blue lines use a Clayton copula. Lines present the copula isolines and dots show observed data. The vertical axis on the right-hand side shows the joint probability value of isolines.**

### 3.2. Contribution of storm surge, astronomical tide and RSLR to peak water level

Figure 6 presents the scatter plot of peak water level and accumulated rainfall with and without RSLR. It shows that the influence of RSLR pushes up the design value of peak water level from 3.25 m to 3.36 m under the 10-yr joint return period, with the corresponding design value of rainfall at 90.39 mm. The univariate analysis approach is to assume independence

between rainfall and sea level, then the independence assumption would generally lead to lower design values compared to scenarios from the copula-based method. It usually depends on how one selects the pairs and the statistical model used

(independent/dependent). This is a direct consequence of the difference in the sampling of extreme observations between both approaches.

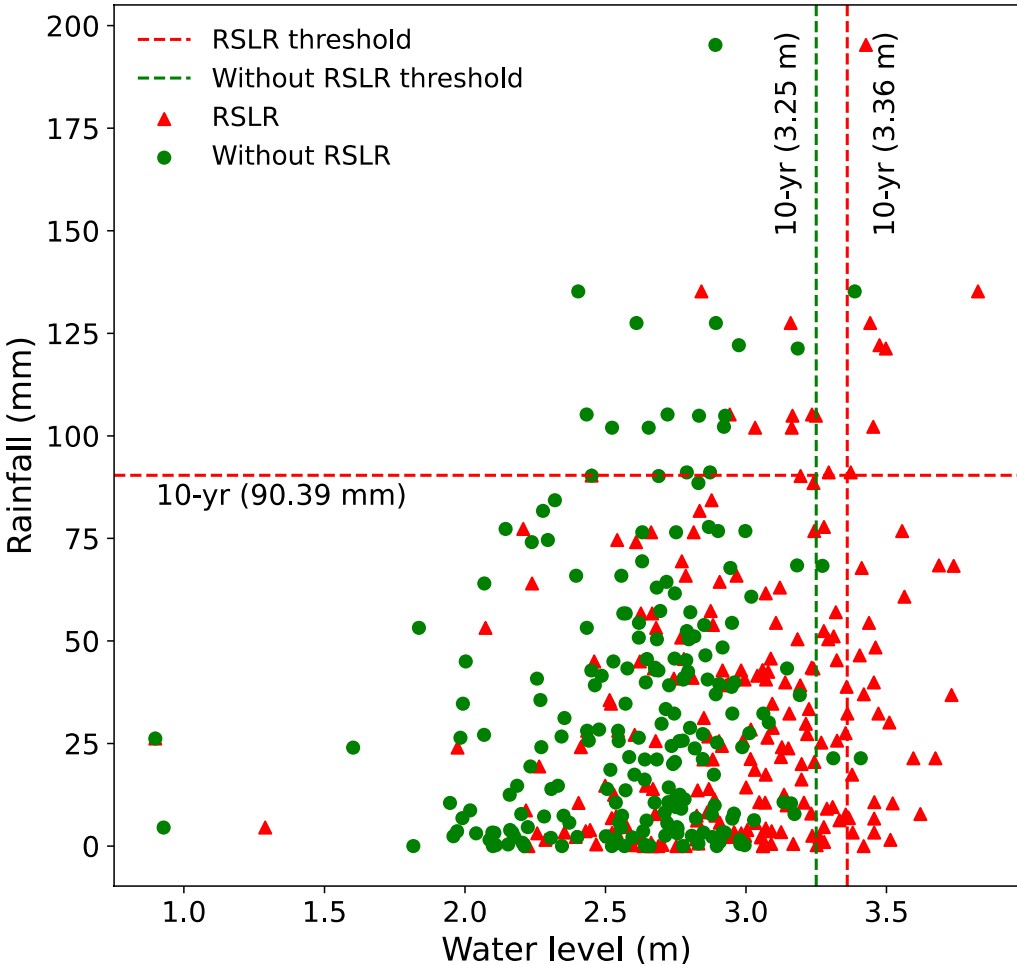

Figure 6. Scatter plot of water level and rainfall analysed. Red lines show the design value of 10-yr joint return period with effect of RSLR. Green lines show the design value of 10-yr joint return period without effect of RSLR.

Based on the results in Figure 6, we defined the compound flood events as the peak water level and accumulated rainfall both being greater than their design values of 10-yr joint return period (i.e., peak water level > 3.36 m and accumulated rainfall > 90.39 mm). Based on this criterion, we identified seven compound flood events under the influence of RSLR (Figure 7).

Peak water level results from the combination of storm surge, astronomical tide and RSLR. Figure 7 shows the contribution of storm surge, astronomical tide and RSLR to peak water level from the seven extreme compound flood events in Shanghai. We consider the cases including the effect of RSLR and split the peak water level according to the contributions of its components, i.e., storm surge, astronomical tide, and RSLR, to investigate their shares of contribution.

Overall, storm surge explains 32% of the peak water level, while astronomical tide accounts for 55% and RSLR accounts for 13% of the peak water level. The astronomical tide is in general the leading contributor to the peak water level, but storm surge can be the leading contributor in some cases, e.g., TC 4, in which the contribution of storm surge accounted for 45% of peak water level. Under the scenario of future global warming and further urbanization, the impact of RSLR would increase and should not be treated as less important.

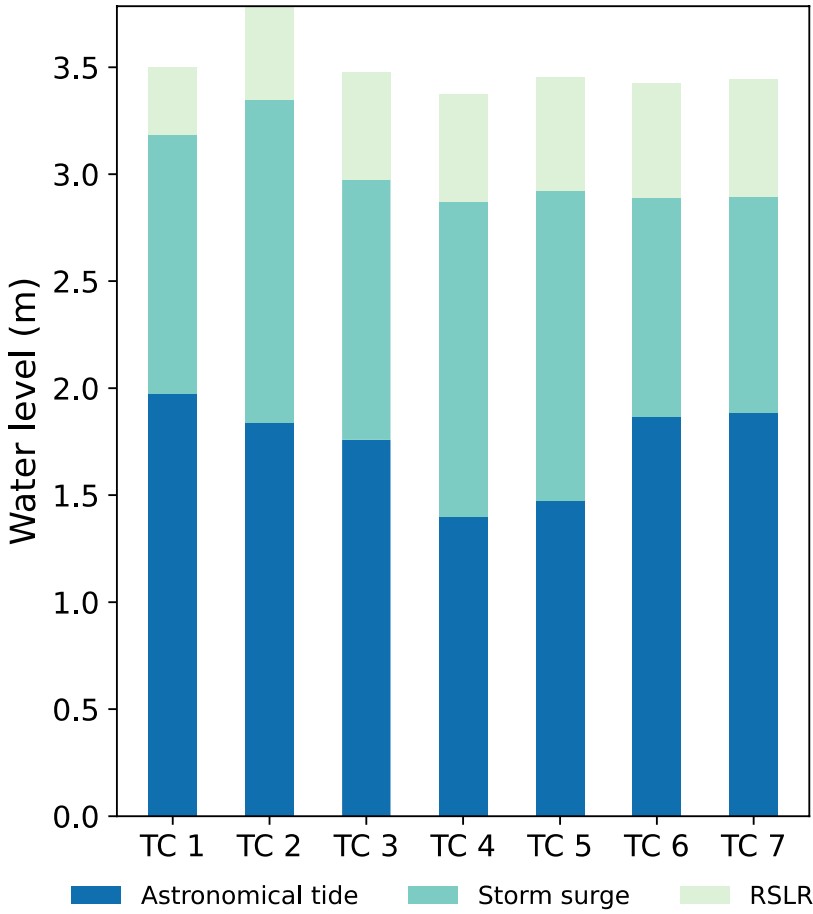

**Figure 7. The contribution of storm surge, astronomical tide and RSLR to peak water level. (Cases are samples greater than 10-yr joint return period.)**

**4 Discussion**

    Coastal areas are the most densely populated and economically developed areas of many countries, and they are also the
285 most vulnerable regions to the risk of compound floods from heavy precipitation and storm surge because of high population and property density as well as storm surge risk (Shen et al., 2019). In this study, we provide a framework that could be applied in general to coastal cities which face the constraint of unavailable water level records.

## 4.1 The dependency between the water level and rainfall

The dependence among different drivers of compound floods has been widely studied. For example, Zheng et al. (2013) identified significant dependence between precipitation and storm surge along the coastlines of Australia; Wahl et al. (2015) examined the enhanced dependence between precipitation and storm surge, and reported an increasing trend in compound flood risk in past decades along the coast of the US. These findings are critical to better understand the changing compound flood risk and provide important references for the evaluation of simulation-based studies.

The correlations between rainfall and storm surge are determined by various factors such as meteorological conditions and regional topography. For example, compound floods from heavy precipitation and storm surge can occur during TCs (Wahl et al., 2015; Bevacqua et al., 2019). TCs are one of the most important triggers of compound floods from heavy rainfall and storm surge in coastal regions. Even though compound floods are receiving attention, few studies have analyzed the dependency between water level and rainfall during historical TCs in China. This study enriches this stream of literature by quantifying the joint distribution of peak water level and rainfall during TCs in the Shanghai estuary region. On the other hand, it is worth noting that the record lengths of observational data in our study is relatively short and the uncertainties of simulation-based studies could be large. Therefore, further studies are needed once more observational data become available.

## 4.2 The effect of RSLR on peak water level

Deltas are especially vulnerable to RSLR because of their low elevation and commonly high rates of land subsidence (Wang et al., 2012; Higgins et al., 2014). Long-term tide gauge records show that global mean sea levels have risen by $1.7 \pm 0.3$ mm/yr over the last century (Holgate, 2007; Cipollini et al., 2017). Nearly 90% of the world's river deltas suffer the impact of RSLR, including Shanghai and Manila (He and Silliman, 2019). The accelerated rise of global sea level puts low-lying coastal regions at risk of increases in frequency and magnitude of flooding (Cazenave and Cozannet, 2014). For example, the sea level rose on average by $\sim$10 cm over the 20th century along the Italian coast, and flood frequency increased by more than seven times in the area (Kulp and strauss, 2019). Increased flooding because of RSLR, in regions that experience storm surges from TCs, further increases the vulnerability of coastal regions to inundation (Edmonds et al., 2020).

Previous studies of Shanghai reported increased risk of coastal floods due to global and local changes (Wang et al., 2012; Yin et al., 2013; Yan et al., 2016). Including the increased RSLR we estimated over the past 58 years (0.55 m), a 4.3 m projected RSLR due to additional land subsidence along the Yangtze River delta by 2100 would result in half of Shanghai being flooded by extreme storm-water levels (Wang et al., 2012). There are serval potential carbon emission scenarios used to project sea-level rise. Regardless of the methods and emission scenarios used to estimate future sea levels, the consensus is that sea levels are rising and its rate is expected to accelerate (Wahl et al., 2017). By contrast, this paper presents a probabilistic analysis of the impact of RSLR on peak water level, accounting for the effects of sea level rise and land

subsidence on coastal flooding, in Shanghai over 1961-2018. The findings from our research would provide more solid foundation for the scenario-based analysis towards future and be useful for the decision-making about the adaption via coastal flood defense measures for Shanghai.

### 4.3 Multiple contributors to peak water level

Coastal flooding from peak water levels is one of the most devastating natural hazards to Shanghai. A storm with strong
winds and low atmospheric pressure can produce a large storm surge and large waves. A storm surge is an increase in water level above normal sea level and is a function of storm intensity, duration, size, and location (Cooper et al. 2008). Tides are an astronomical phenomenon caused by the gravitational attraction of the moon and the sun on earth's oceans, while storm surge is a meteorological phenomenon (Karim and Mimura 2008). If storm surge coincides with the astronomical high tide, these water levels superpose, and an extreme water level may be generated. Southeast Asia is highly vulnerable to, and
frequently impacted by, extreme sea-level events of different origins: TCs cause severe storm surges and rainfall with potentially devastating impacts to the economy and environment and in many cases loss of human life.

Astronomical tides are deterministic and can be predicted far in advance, whereas storm surges can only be accurately hindcast from tide gauge records. Prediction of storm surge is possible days in advance of TC landfall, simulated by taking into account predicted forcing variables, such as wind stress and sea level pressure over the sea surface. Tide gauge records
have been used to study sea level extremes. However, 90% of the tide gauges located in Southeast Asia have record lengths of less than 30 years. One way to overcome the absence of long tide gauge records is to employ numerical models to simulate the storm surge component (Park and Suh, 2012) using best track TC data or meteorological reanalysis results, as we have done in this research.

In this study, we demonstrate that the astronomical tide plays an important role in the total water level in Shanghai. Indeed,
surges might occur at any tidal level, and are especially strong in shallow estuaries. A high tide at Wusongkou gauge would extend to downtown Shanghai causing a fluvial flood. The flood extent, depth, and duration can be exacerbated by storm surge, and consequently, the disruptive impact increases strongly (Ke et al., 2018). Astronomical tides contribute to peak water levels during TCs (Sweet et al., 2009). Bacopoulos (2017) showed that in the St Johns River in Florida, the astronomical tide could contribute as much as 94% to the extreme water level. Our study highlights that the critical
components to consider in the analysis of peak water level during TCs are the astronomic tides, storm surge and RSLR. In future research, we will explore the applicability of the presented methodology to other regions where limited observational data availability has hampered a better understanding of peak water level, storm surge and potential changes related to climate variability and change.

### 5 Conclusions

It is important to consider the compounding effects of multiple interdependent extremes or drivers to accurately characterize the underlying hazard. In this study, we focused on the joint impact of peak water level and accumulated rainfall in

Shanghai, a coastal mega-city located in the Yangtze River Delta region. We showed that Shanghai is prone to compound flooding and this justifies the adoption of a probabilistic modelling framework to incorporate the interdependence of multiple flood-drivers.

Between 1961 and 2018, the RSLR had increased by 0.55 m in Shanghai. With the ongoing global warming and further urbanization vertically and horizontally in the city, the process of RSLR would continue and amplify the peak water levels in extreme flooding events. The sample data we consolidated show an increase in the probability of peak water level under the effect of RSLR. We also identify the extent of the shift in the joint distribution of peak water level and accumulated rainfall during TC periods between the theoretical setting without RSLR and the real setting with RSLR by employing the best fitted

copula functions. The shift indicates that the RSLR leads to an increase in both the mean and variance of peak water levels, thus a significantly higher level of flooding risk in Shanghai.

The design value of peak water level and accumulated rainfall are 3.36 m and 90.39 mm during TCs under the 10-yr joint return period and with the influence of RSLR. We selected the potential compound flood events according to this pair of design value and identified seven potential compound flood events. The analysis of these seven events shows that astronomic

tide is in general the most important driver of the peak water level, however, there is one case in which storm surge is the leading driver of the peak water level. If the astronomic tide is relative to mean high water instead of mean sea level, the length of the tide part bars may be smaller. However, we argue that the peak water level is the most dangerous hazard to coastal cities. The combination of astronomical tide, storm surge and RSLR drives the peak water level. We cannot neglect the contribution of tide during the typhoon season.

The framework developed in this study could be applied to other coastal cities or regions in East and Southeast Asia. The impact of the RSLR in amplifying the peak water level would significantly increase in the future. Therefore, the monitoring and prediction of the RSLR should be an important component in the development of future design standards for flood preparedness. Furthermore, RSLR caused by climate change and intensive use of urban land would also increase social vulnerability, which can be an interesting topic for future research.

**Data availability**

IBTrACS data were obtained from the NOAA National Centers for Environmental Information from their website at https://www.ncdc.noaa.gov/ibtracs. Daily rainfall records from 1961 to 2018 are collected from the China Meteorological Administration (CMA, http://data.cma.cn).

**Competing interests**

The authors declare that they have no conflict of interest.

## Acknowledgments

This work is sponsored by the National Key R&D Program of China (Grant no. 2019YFE0124800); National Natural Science Foundation of China (Grant number: 51761135024 and 41971199), Netherlands Organization for Scientific Research (NWO) (Grant no. ALWSD.2016.007), the Engineering and Physical Sciences Research Council of the UK (Grant Nos. R034214/1); and also the Shanghai Science and Technology Support Program (Grant No. 19DZ1201505), the Major Program of National Social Science Foundation of China (Grant No. 18ZDA105), and the ECNU Academic Innovation Promotion Program for Excellent Doctoral Students (YBNLTS2020035). the High-level Special Funding of the Southern University of Science and Technology (Grant No. G02296302, G02296402). Thankful for financial support from the program of China Scholarships Council (No.202006140040).

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
