# Peer review of "Compound flood impact of water level and rainfall during tropical cyclone period in a coastal city: The case of Shanghai"

_Natural Hazards and Earth System Sciences, 2022_

## Author Comment (AC1)

We sincerely thank Referee #1 for his/her careful review and constructive comments. We truly believe that the changes suggested by Referee #1 would enhance the quality of the manuscript. A point-by-point response is presented in Supplements.

**1. please give more clarification on motivation of writing this paper, i.e. significant importance in Shanghai.**

Thanks for your comments. Shanghai is the largest and most developed coastal megacity in China. Shanghai is surrounded by water on three sides, and the Huangpu River passes through the city. Rainstorm and storm surge caused by typhoon from June to October often cause substantial losses. For example, Typhoon Winnie killed seven people and flooded more than 5,000 households in 1997. Although the construction of flood control measures in the past 50 years has effectively reduced the risk of storm floods, Typhoon Matsa in 2005, Typhoon Fitow in 2013, and Typhoon Lekima in 2019 brought significant damage to Shanghai. However, owing to the unavailability of water level records during typhoon events, there is no research that has been able to calculate the joint probability of peak water level and rainfall during historical tropical cyclones (TCs) and to calculate the marginal contribution of each driver as we have done in this research. Our modeling framework couples a state-of-the-art hydrodynamic model and statistic model. This model coupling enables us to quantify the joint distribution of peak water level and rainfall during typhoon season, and also to consider the comparative cases with and without RSLR for Shanghai. This framework could be applied to other coastal cities which face the similar constraint of unavailable water level records. This is the major motivation and contribution of our research.

We rewrite paragraph 2 in the Introduction to emphasize the significant importance of compound flood risk in Shanghai.

"Coastal regions are usually the most densely populated and economically developed areas of a country, and they are also the most vulnerable regions to the risk of compound flooding from heavy rainfall and extreme storm surge due to this large population and property density (Neumann et al., 2015). Shanghai is the largest and most developed coastal megacity in China. Rainstorm and storm surge caused by typhoon from June to October often cause substantial losses (Li et al., 2018; Yin et al., 2021). For example, extreme storm flooding caused nearly 30 thousand casualties in 1905 (Li et al., 2019). In 1962, storm flooding inundated half of the downtown region for nearly 10 days due to 46 failures along the floodwalls of the Huangpu River and its branches and led to huge losses of 1/6 of the local Gross Domestic Product (GDP) in Shanghai (Ke, 2014). In 1997, Typhoon Winnie killed seven people and flooded more than 5,000 households due to the extreme storm surge and rainfall (Ke et al., 2021). Although the construction of flood control measures in the past 50 years (especially after typhoon Winnie in 1997) has effectively reduced the risk of storm surge and rainstorm floods, Typhoon Matsa in 2005 (US $2.23 billion damage), Typhoon Fitow in 2013 (US $10.4 billion damage), and Typhoon Lekima in 2019 (US $2.55 billion damage) also brought significant damage to Shanghai (Du et al., 2020). Given the substantial damage caused by compound flooding, comparing the encounters of rainfall and storm surge during typhoon season is urgent in order to understand the driving mechanisms and frequency of compound flooding in Shanghai. However, owing to the unavailability of water level records, there is little research that has been able to estimate the dependency between peak water level and accumulated rainfall during historical TCs."

**2. Please add the discussion in the context to support your findings with references.**

Thank you for your suggestion. We have added a new discussion section to support our findings with additional references. In this new section, we discussed the dependency between the water level and rainfall, the effect of RSLR on peak water level and the multiple contributors to peak water level. These discussions further convince the feasibility of the probabilistic modelling framework we developed in incorporating the interdependence of multiple drivers. We highlighted our key finding that the peak water level is the most dangerous hazard to Shanghai. The combination of astronomical tide, storm surge and RSLR drives the peak water level. In the future research, it is essential to take into account the contribution of tide during the typhoon events. We expect the findings from our research to be useful for the decision-making on the adaption via coastal flood defense measures for Shanghai and other coastal cities or regions in East and Southeast Asia. The new discussion section is as follows.4 Discussion

[revised manuscript text omitted]

In this study, we argue that the astronomical tide plays a role in the total water level in Shanghai. Indeed, surges might occur at any tidal level, and are especially strong in shallow estuaries. A high tide at Wusongkou gauge would extend to downtown Shanghai causing a fluvial flood. The flood extent, depth, and duration can be exacerbated by storm surge, and consequently, the disruptive impact increases strongly (Ke et al., 2018). Astronomical tides contribute to peak water levels during TCs (Sweet et al., 2009). 75% of extreme storm surges coincided with astronomical high tide, where the astronomical tides contribute 94% of the water level (Bacopoulos, 2017). The critical components to consider in the analysis of peak water level during TCs are the astronomic tides, storm surge and RSLR . In future research, we will explore the applicability of the presented methodology to other regions where limited observational data availability has hampered a better understanding of peak water level, storm surge and potential changes related to climate variability and change.

---

## Author Comment (AC2)

We sincerely thank Referee #2 for his/her careful review and valuable advice. Based on the comments and suggestions, we have made extensive revisions to the original manuscript. A point-by-point response is presented in Supplements.

**1. The storyline of introduction can be improved for enhancing the readability. The significance of this work could be increased by discussing the results in a wider sense.**

Thanks for your comments, we revised the introduction, and added discussion section to support our findings.

**2. L35-38: a relatively long sentence.**

We rewrite this long sentence. "As such, the joint probability theory has been incorporated into the analysis of compound flood risk to take the advantage of the Sklar's Theorem (M. Sklar, 1959). According to Sklar's Theorem, any multivariate joint cumulative distribution function can be expressed in terms of univariate marginal distribution functions and a copula which describes the structure of dependency between the variables (Bevacqua et al., 2019)."

**3. L43: what does GDP stand for?**

The "GDP" means the Gross domestic product, which is a measurement of total value-added produced with a region. We now spell it out in the manuscript: "the local Gross Domestic Product (GDP) in Shanghai".

**4. L46-47: could the authors please provide a rough estimate of the damage due to each Typhoon in US dollars? (I saw such numbers in Section 2.1)**

Thanks for your comments. We provided rough estimation of the economic damage caused by each of these typhoon events. This sentence is changed to "Although the construction of flood control measures in the past 50 years (especially after the typhoon Winnie in 1997) has effectively reduced the risk of storm surge and rainstorm floods, Typhoon Matsa in 2005 (US $2.23 billion damage), Typhoon Fitow in 2013 (US $10.4 billion damage), and Typhoon Lekima in 2019 (US $2.55 billion damage) also brought significant damage to Shanghai (Du et al., 2020)".

**5. L48-49: Please double check the grammar of the sentence.**

Thanks for your comments. To improve the readability of this paper, we changed this sentence to "However, owing to the unavailability of water level records, there is little research that has been able to estimate the dependency between peak water level and accumulated rainfall during historical TCs.".

**6. L106: greater -> higher**

Thanks for noting this. We changed "greater" to "higher".

**7. L125: costly -> severe**

Thanks for noting this. We changed "costly" to "severe".

**8. L230: I would expect one or two sentences for describing the results of Figure 5.**

Thanks for your comments. We added the following description at the end of this paragraph: "Figure 5 shows the difference between peak water level and accumulated rainfall with RSLR and without RSLR. This indicates that different copula families can return different dependence structures. In Figure 5, both peak water level and accumulated rainfall are presented in probability space. Gaussian and Clayton copula families are used to explain the bivariate dependence between peak water level and accumulated rainfall in this study. The red and blue isolines are fitted Gaussian copulas and Clayton copulas, respectively. Neither is among the commonly used copulas in the hydrological literature. This highlights the importance of the choice of the copula, and quantifies the difference in results based on copula choice."

**9. L239: What do the authors refer to with the traditional approach?**

Thanks for your comments. The "traditional approach" means the univariate statistics, which characterizes one variable only. We now directly call it univariate analysis approach. The revised sentence reads as follows: "The univariate analysis approach is to assume independence between rainfall and sea level, then the independence assumption would generally lead to lower design values compared to scenarios from the copula-based method."

**10. L254: account -> accounts**

Thanks for noting this. It has been corrected.

---

## Author Response (AR1)

**General comments:**

The authors investigate compound flood potential from historical TCs impacting the Shanghai region of China. They model TC-induced storm tides (surge + astronomical tide) using the Delft3D hydrodynamic model and utilize observed rainfall from nearby tidal gauges. They quantify the joint probability of rainfall and sea level using a bivariate copula, and then assess the impact of historical relative sea level rise (RSLR). The results show that the astronomic tide is the lead driver of the peak coastal water level, followed by the impact of storm surge. Relative sea level rise also significantly amplified the peak coastal water level in the study period of 1961-2018. This paper is on a topic of interest to the audience. The modeling and analysis methods are scientifically sound. The results provide helpful insights about coastal compound floods. I have a few comments that I hope the authors could address in their revision:

**Overall Response**: We would like to thank these detailed comments and suggestions, which are very helpful for us to improve the quality of the manuscript. In the revision, we have taken into consideration all suggestions and addressed all concerns raised. In this letter we report the point-by-point response to the comments.

**Specific comments:**

**1. Line 20: Please change "Delft3D-Flow Flexible Mesh" to "D-Flow Flexible Mesh". To my knowledge, the Delft3D Flexible Mesh is suite software and the D-Flow Flexible Mesh is the main module of this suite.**

**Response**: Thank you for your suggestion. We have changed "Delft3D-Flow Flexible Mesh" to "D-Flow Flexible Mesh". In section 2.4, we rephase the sentence as "The D-Flow FM module as part of the Delft3D Flexible Mesh suite solves the non-linear shallow water equations for unsteady flow and transport phenomena derived from the three-dimensional Navier Stokes equations for incompressible free surface flow (Symonds et al., 2016)."

References:
Symonds, A. M., Vijverberg, T., Post, S., Van Der Spek, B. J., Henrotte, J., Sokolewicz, M. (2016). Comparison between Mike 21 FM, Delft3D and Delft3D FM flow models of western port bay, Australia. Coast. Eng., 2, 1-12.

**2. Line 21: Worth mentioning "historical" relative sea level rise as it is not immediately clear if this work is examining historical compound flooding or projecting future compound flooding.**

**Response**: We employed historical TCs in this study and highlighted them in the abstract and rephrased this sentence to "This study employed the D-Flow Flexible Mesh model to simulate the historical peak coastal water level, consisting of storm surge, astronomical tide, and the relative sea level rise (RSLR) in Shanghai over 1961-2018."

**3. Lines 35-38: The sentence beginning " As such, the joint probability theory..." does not make sense. Please rephrase.**

**Response**: We rewrite this long sentence as follows. "As such, the joint probability theory has been incorporated into the analysis of compound flood risk to take the advantage of Sklar's Theorem (M. Sklar, 1959). According to Sklar's Theorem, any multivariate joint cumulative distribution function can be expressed in terms of univariate marginal distribution functions and a copula which describes the structure of dependency between the variables (Bevacqua et al., 2019)."


**Response**: In section 2.1, we provided more information about the hazards in Shanghai: "The total area of Shanghai is 6,340.5 km$^2$ with a population of 24.87 million in 2020. The annual rainfall is around 1,200 mm. June to September are the rainy months. From late August till early September, Shanghai is frequently affected by typhoons and rainstorms (Yin et al., 2021). Storm flooding caused by typhoons is the main natural hazard in Shanghai. Shanghai's flood risk is about US $63 million/year under an optimistic scenario of a maximum protection level of 1/1000 per year (Hallegatte et al., 2013). Although the construction of flood control measures in the past 50 years has effectively reduced the risk of storm floods, TC Matsa in 2005, the 2013 TC Fitow, and the 2019 TC Lekima caused substantial losses in Shanghai. Particularly, Typhoon Winnie in 1997 led to direct economic damage of over US $100 million. During typhoon Winnie, the peak water level at Huangpu Park (city center) rose to 5.72 m, equivalent to the water level with a 500-year return period. During Typhoon Fitow in 2013, the water level at Mishidu in the inland area of the Huangpu River was recorded at WD (Wusong Datum is adopted as the reference) as 4.61 m, the highest on record (Ke et al., 2018). In the context of climate change, relative sea level rise, and urban expansion, Shanghai will face higher compound flood risk and challenges from TCs, storm surge, and extreme rainstorms in the future (Wang et al., 2018)."

In this study, we employed the Holland model to reconstruct the TCs in the past, with the Takagi and Wu (2016) empirical relation used to determine radius of maximum winds where this information was missing in the best track data. Delft3D WES (Wind Enhance Scheme), a built-in module in Delft3D, is used in this study to generate wind fields of TC scenarios. WES calculates the wind and pressure according to the Holland formula (Holland et al., 2010). It is able to generate tropical cyclone wind and pressure fields around storm center positions on a high-resolution grid. This asymmetry is caused by the use of the translational speed of the cyclone's center displacement as the steering flow, and the rotation of the wind velocity due to friction (Takagi and Wu, 2016). This model performance has been assessed based on model configuration, model calibration, grid generation and computational efficiency (Ke et al., 2019). The output of WES is suitable as input for the Delft3D-Flow model to simulate water level including the effect of storm surge.

In addition, we have added a map to show the tracks of historical tropical cyclones used in this study (Figure 2). We hope that this could help readers better understand the TCs hazards in Shanghai in the past 60 years.

[Figure]

Figure 2. Location map for the area of interest. (Grey colored lines indicate major historical typhoon tracks within the region. Blue box indicates the selection criteria.)

**Response**: Thanks for your comments. We have rephased the figure's caption as follows, "With RSLR (red) and without RSLR (blue) for 2 different copulas. Both peak water level (x axis) and accumulated rainfall (y axis) are presented in probability space. The red isolines are the fitted Gaussian copula, and the blue lines use a Clayton copula. Lines present the copula isolines and dots show observed data. The vertical axis on the right-hand side shows the joint probability value of isolines". By the way, we also added the following description at the end of this paragraph: "Figure 5 shows the difference between peak water level and accumulated rainfall with RSLR and without RSLR. This indicates that different copula families can return different dependence structures. In Figure 5, both peak water level and accumulated rainfall are presented in probability space. Gaussian and Clayton copula families are used to explain the bivariate dependence between peak water level and accumulated rainfall in this study. The red and blue isolines are fitted Gaussian copulas and Clayton copulas, respectively. Neither is among the commonly

used copulas in the hydrological literature. This highlights the importance of the choice of the copula, and quantifies the difference in results based on copula choice."

[Figure]

Figure 5: With RSLR (red) and without RSLR (blue) for 2 different copulas. Both peak water level (x axis) and accumulated rainfall (y axis) are presented in probability space. The red isolines are the fitted Gaussian copula, and the blue lines use a Clayton copula. Lines present the copula isolines and dots show observed data. The vertical axis on the right-hand side shows the joint probability value of isolines.

**14. Please provide the summary of the significant findings in discussion part, for example, the dependency of water level and rainfall. In discussion part, the authors should make a comparison and contrast of the findings with others.**

**Response**: Thank you for your suggestion. We have added a new discussion section to support our findings with additional references. In this new section, we discussed the dependency between the water level and rainfall, the effect of RSLR on peak water level and the multiple contributors to peak water level. We compared our findings with other researches which reported parallel results. For example, the literature shows that the correlation between rainfall and surges is generally weak, our research highlights that the correlation has a significant impact on coastal floods. Another example is that in the St Johns River in Florida, the astronomical tide contributes as much as 94% to the extreme water level (Bacopoulos, 2017). These discussions further convince the feasibility of the probabilistic modelling framework we developed in incorporating the interdependence of multiple drivers. We highlighted our key finding that the peak water level is the most dangerous hazard to Shanghai. The combination of astronomical tide, storm surge and RSLR drives the peak water level. In future research, it is essential to take into account the contribution of the tide during TCs. We expect the findings from our research to be useful for the decision-making on 
[revised manuscript text omitted]